# Electric-field tunable Type-I to Type-II band alignment transition in MoSe$_2$/WS$_2$ heterobilayers

Jed Kistner-Morris[1,6], Ao Shi[1,6], Erfu Liu [1,2], Trevor Arp[1,3], Farima Farahmand [1], Takashi Taniguchi [4], Kenji Watanabe [5], Vivek Aji[1], Chun Hung Lui [1] ✉ & Nathaniel Gabor [1] ✉

Semiconductor heterojunctions are ubiquitous components of modern electronics. Their properties depend crucially on the band alignment at the interface, which may exhibit straddling gap (type-I), staggered gap (type-II) or broken gap (type-III). The distinct characteristics and applications associated with each alignment make it highly desirable to switch between them within a single material. Here we demonstrate an electrically tunable transition between type-I and type-II band alignments in MoSe$_2$/WS$_2$ heterobilayers by investigating their luminescence and photocurrent characteristics. In their intrinsic state, these heterobilayers exhibit a type-I band alignment, resulting in the dominant intralayer exciton luminescence from MoSe$_2$. However, the application of a strong interlayer electric field induces a transition to a type-II band alignment, leading to pronounced interlayer exciton luminescence. Furthermore, the formation of the interlayer exciton state traps free carriers at the interface, leading to the suppression of interlayer photocurrent and highly nonlinear photocurrent-voltage characteristics. This breakthrough in electrical band alignment control, interlayer exciton manipulation, and carrier trapping heralds a new era of versatile optical and (opto)electronic devices composed of van der Waals heterostructures.

Heterojunctions, where two different materials meet, form the fundamental building blocks for modern functional devices[1], such as light emitting diodes[2], photodetectors[3] and field-effect transistors[4]. The critical determinant of heterojunction device characteristics lies in the alignment of conduction and valence bands between the two materials[5–7]. Type-I band alignment occurs when the band gap of one material is fully encompassed within the band gap of the other material, i.e., both the conduction band minimum (CBM) and the valence band maximum (VBM) of the heterostructure reside in the same material. This straddled band alignment causes photoexcited electrons and holes to relax into the same medium[8,9] (Fig. 1a, b). As a result, the excitons have larger electron-hole wavefunction overlap, higher oscillator strength, and shorter radiation lifetime, which favor applications in light emitting devices[10,11]. In contrast, type-II band alignment situates the CBM of the heterostructure in one material and the VBM in the other (Fig. 1c, d). This staggered alignment causes the photoexcited electrons and holes to relax to different materials. This facilitates exciton dissociation and

[1]Department of Physics and Astronomy, University of California, Riverside, CA 92521, USA. [2]National Laboratory of Solid State Microstructures, School of Physics, and Collaborative Innovation Center of Advanced Microstructures, Nanjing University, Nanjing 210093, China. [3]Department of Physics, University of California, Santa Barbara, CA 93106, USA. [4]International Center for Materials Nanoarchitectonics, National Institute for Materials Science, 1-1 Namiki, Tsukuba 305-0044, Japan. [5]Research Center for Functional Materials, National Institute for Materials Science, 1-1 Namiki, Tsukuba 305-0044, Japan. [6]These authors contributed equally: Jed Kistner-Morris, Ao Shi. ✉e-mail: joshua.lui@ucr.edu; nathaniel.gabor@ucr.edu

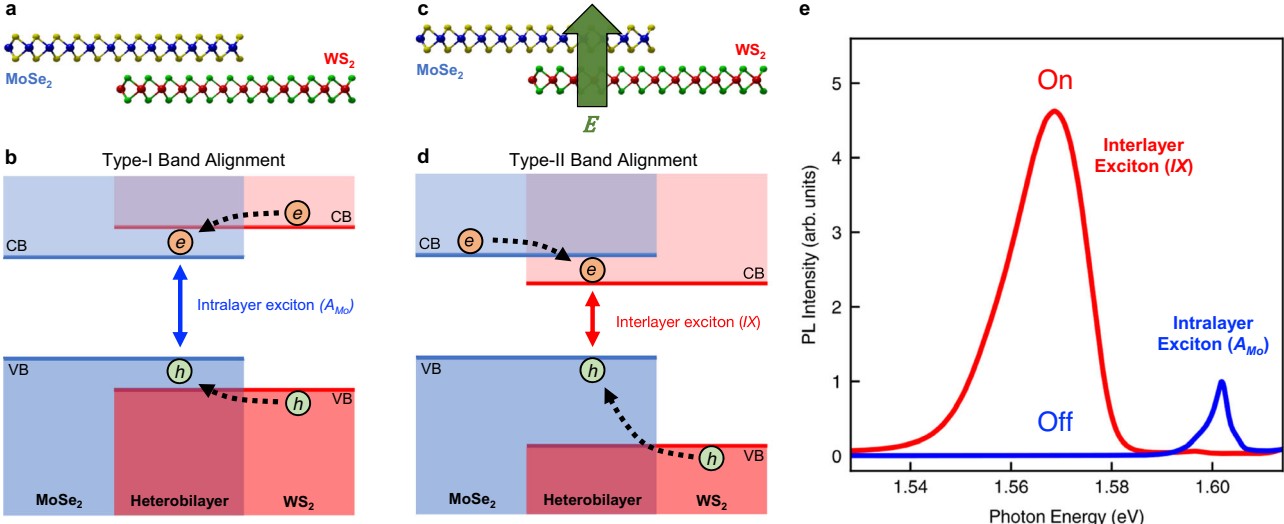

**Fig. 1 | Illustration of electric-field-induced type-I to type-II band alignment transition in MoSe₂/WS₂ heterobilayer. a**, **b** Atomic layers and energy band diagram of a MoSe₂/WS₂ heterobilayer with no electric field. The position-dependent conduction bands (CB) and valence bands (VB) of monolayer MoSe₂ (blue) and WS₂ (red) exhibit the type-I alignment. The photoexcited electrons and holes relax to the MoSe₂ layer to form intralayer excitons ($A_{Mo}$). **c**, **d** Atomic layers and energy band diagram under strong vertical electric field. The heterobilayer exhibits type-II band alignment. The photoexcited electrons (holes) relax to the WSe₂ (MoSe₂) layer to form interlayer excitons ($IX$). **e** Measured photoluminescence (PL) spectra under zero (blue) and $E = 0.25$ V/nm (red) electric field. The interlayer excitons are turned on and off by the electric field.

photocarrier extraction and hence favors applications in photodetection[12,13] and photocatalysis[14].

In the conventional scenario, the band energy offset between two materials is fixed; this imposes many constraints on device functionality. To usher in the next generation of devices, it is highly desirable to transition between different alignment types within a single heterostructure. The realization of tunable band alignment holds immense potential for unlocking unprecedented multifunctionality in device applications. However, achieving such demonstrations is challenging. While certain approaches involve material engineering through adjusting chemical compositions[15–18] or layer thickness[19], these methods necessitate multiple synthesis iterations, precise dopant control, or the use of different samples. A more efficient alternative envisions the capability to tune the band alignment types through straightforward electrical means in a single device, though, to the best of our knowledge, such a demonstration has not been realized.

A groundbreaking advancement in the fabrication of band-engineered heterostructures has emerged within the realm of two-dimensional (2D) van der Waals (vdW) materials[20–24], notably including monolayer transition metal dichalcogenides (TMDs) like MoSe₂ and WS₂. These TMDs feature direct bandgaps, heavy carriers, robust excitons[25–32], and innovative spin-valley-coupled physics[33–37]. In contrast to conventional grown heterostructures, 2D vdW heterostructures exhibit reduced dimensions, atomically sharp interfaces without dangling bonds, and high tolerance for lattice mismatch. These attributes greatly enhance their stability, tunability and overall functionality. Moreover, the continually expanding library of diverse 2D materials offers a rich palette of options for creating vdW heterostructures. Indeed, recent theoretical propositions have delved into the possibility of tuning band alignments in 2D vdW heterostructures through various means, including electric field[38,39], strain[38], interlayer spacing[38], and twist angle[40]. However, experimental demonstration of such band alignment tuning is still lacking.

In this Article, we demonstrate an electric-field-induced transition between type-I and type-II band alignments in MoSe₂/WS₂ heterobilayers, as evidenced through photoluminescence (PL) and photocurrent (PC) spectroscopy. Figure 1 illustrates our key

findings. Initially, in the absence of an external electric field, the heterojunction of monolayer MoSe₂ and WS₂ exhibits a type-I band alignment. In this state, the WS₂ CBM is marginally higher than the MoSe₂ CBM, while the WS₂ VBM is considerably lower than the MoSe₂ VBM (Fig. 1a, b). This configuration, with its straddled band gap, facilitates the relaxation of photo-excited electrons and holes to the same (MoSe₂) layer, leading to a pronounced PL peak of intra-layer excitons ($A_{Mo}$) at 1.6 eV (blue line in Fig. 1e). However, the application of a strong vertical electric field, directed from the WS₂ to MoSe₂ layer, triggers a critical shift—the WS₂ CBM moves below the MoSe₂ CBM (Fig. 1c, d). This alteration leads to a type-II band alignment with staggered band gaps, where the photoexcited electrons and holes tend to relax to different layers—electrons to the WS₂ layer and holes to the MoSe₂ layer. This separation leads to the formation of interlayer excitons ($IX$), which have a lower energy than their intralayer counterpart, resulting in a dominant PL peak at ~1.57 eV (Fig. 1e). Beyond the striking shift of the PL spectrum, the emergence of interlayer excitons also effectively traps electron-hole pairs at the heterojunction, thereby suppressing the interlayer photocurrent ($I_{pc}$). This results in a highly nonlinear relationship between $I_{pc}$ and the applied interlayer voltage. Overall, our findings highlight the profound impact of the type-I to type-II transition on the optical and optoelectronic properties of devices. The ability to electrically control this transition opens up exciting possibilities for designing innovative multifunctional devices using vdW heterostructures.

## Results and discussion
### Photoluminescence measurements
Our experiment employs dual-gate MoSe₂/WS₂ heterobilayers encapsulated in hexagonal boron nitride (BN)[41] (Supplementary Fig. S1, Fig. 2a). We use thin graphite flakes to contact the TMDs and electrodes to enhance device performance. The heterobilayers for PL measurements have twist angles of either ~0° or ~60°. Deviation from these angles will suppress the interlayer emission due to electron-hole momentum mismatch. Below we will present the PL results of Device 1 while the reflectance contrast results are presented in Section 4 of the Supplementary Information.

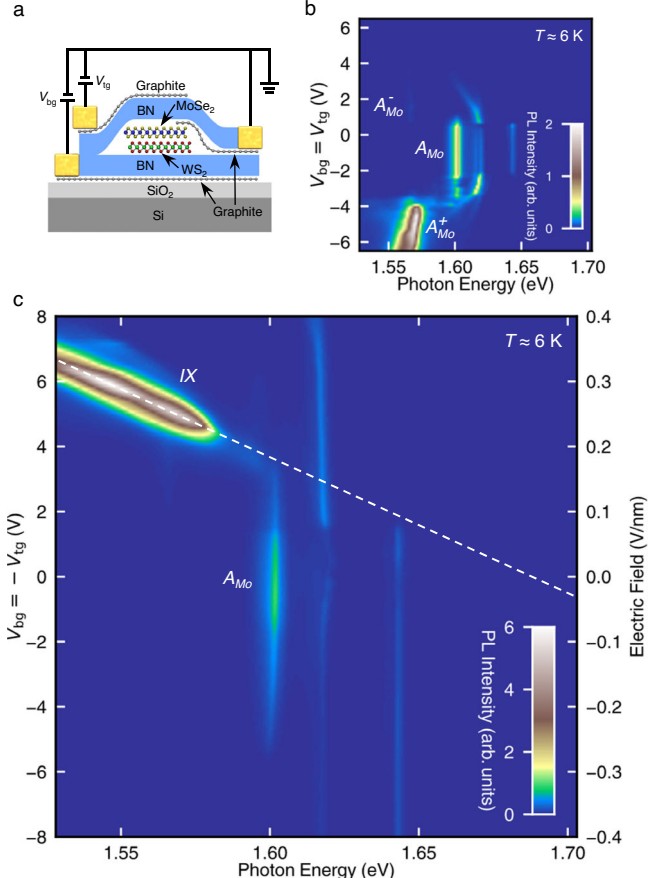

**Fig. 2 | Photoluminescence signature of type-I to type-II band alignment transition in MoSe₂/WS₂ heterobilayers. a** The schematic of a dual-gate MoSe₂/WS₂ heterobilayer device encapsulated by boron nitride. **b** The charge-density-dependent PL map of Device 1. Equal voltages $V_{bg} = V_{tg}$ are applied to the bottom and top gates. The charge density is proportional to the gate voltages. The $A_{Mo}$, $A_{Mo}^-$, $A_{Mo}^+$ features arise from the intralayer excitons, electron-side and hole-side exciton polarons (or trions) in the MoSe₂ layer, respectively. **c** The electric-field-dependent PL map of Device 1. Opposite voltages $V_{bg} = -V_{tg}$ (left axis) are applied to the bottom and top gates to induce an interlayer electric field (right axis). An interlayer exciton (IX) feature appears at high electric field. The dashed line is a linear extrapolation of its Stark shift to zero field. All measurements were performed with 532-nm laser excitation (incident power ≈ 3 μW) at sample temperature $T ≈ 6$ K.

In Device 1, the top and bottom BN have similar thickness. By applying voltages with the same sign on the bottom gate ($V_{bg}$) and top gate ($V_{tg}$), we can inject carriers into the sample without inducing any vertical electric field. The charge-density-dependent PL map of Device 1 (Fig. 2b) exhibits a pronounced line ($A_{Mo}$) at 1.60 eV at the charge neutrality region, which matches the reported A-exciton energy in BN-encapsulated MoSe₂ monolayers[42–45]. Upon injecting electrons or holes into the heterobilayer, the $A_{Mo}$ peak subsides and two new PL peaks ($A_{Mo}^-$, $A_{Mo}^+$) emerge at ~30 meV below the exciton on the electron and hole side, respectively. This energy separation is close to the known exciton-polaron (or trion) binding energies in monolayer MoSe₂[42–45]. Therefore, we infer that $A_{Mo}$, $A_{Mo}^-$, $A_{Mo}^+$ originate from the intralayer exciton and exciton polarons in the MoSe₂ monolayer. We note that the weak lines at higher energies than $A_{Mo}$ may arise from moiré effect or sample inhomogeneities, since they are not reproducible in other devices (Supplementary Figs. S8 and S9). Atomic reconstruction is unlikely to occur in this system due to the small moiré wavelength (~8 nm).

By applying voltages with opposite signs on the bottom and top gate ($V_{bg} = -V_{tg}$), we can apply a vertical electric field between the MoSe₂ and WS₂ layers while maintaining the heterobilayer in the charge neutrality regime. Figure 2c displays the PL map of Device 1 at varying $V_{bg} = -V_{tg}$ (left axis), from which we extract the out-of-plane electric field (right axis) with a BN dielectric constant of 3.4 (see the details in Supplementary Information, Section 2.1). At weak electric field ($E < 0.16$ V/nm pointing from WS₂ to MoSe₂), the $A_{Mo}$ PL line remains pronounced and exhibits no Stark shift; this observation confirms its intralayer nature and supports the type-I alignment of the heterobilayer (Fig. 1b). When the electric field exceeds 0.16 V/nm, the $A_{Mo}$ line subsides and below it emerges a new PL peak (IX). The IX peak redshifts linearly with a slope of 44 ± 11 meV per 0.1 V/nm of field; the ±11 meV error is mainly due to the uncertain BN dielectric constant (2.6–4.2) with a minor ±0.9 meV linear-fit uncertainty. At high field ($E > 0.23$ V/nm), IX becomes bright and dominates the PL.

The Stark shift of IX indicates that it has an out-of-plane dipole, a signature of interlayer excitons. By assuming that the electron and hole are localized in different layers, we deduce an electron-hole separation of 0.4 ± 0.1 nm based on the Stark shift. This separation is comparable to the interlayer spacing (~0.6 nm) of the heterobilayer[46], providing evidence for the origin of interlayer excitons. When we extrapolate the Stark shift of IX linearly to zero electric field, we arrive at an energy of ~1.69 eV, which is ~90 meV above the $A_{Mo}$ line at 1.60 eV. Considering the different exciton binding energies between $A_{Mo}$ and IX, we further estimate that the WS₂ CBM resides ~40 meV above the MoSe₂ CBM, comparable to a predicted 0.03-eV band offset in ref. 47 (see Supplementary Information, Section 3).

Our observation can be readily explained using the schematics in Fig. 1. At low field, the heterobilayer exhibits a type-I alignment with the WS₂ CBM lying slightly above the MoSe₂ CBM. Consequently, photocarriers relax to the MoSe₂ layer to form intralayer excitons (Fig. 1a, b). As the electric field (directed from WS₂ to MoSe₂) increases, the WS₂ CBM is lowered, leading to a transition to a type-II alignment. In the type-II configuration, photoexcited electrons and holes relax to different layers to form interlayer excitons (Fig. 1c, d). We note that an opposite electric field (from MoSe₂ to WS₂) elevates the WS₂ CBM and does not induce the type-II alignment transition. This is consistent with our observation that no IX peak appears at negative electric field (Fig. 2c).

## Photocurrent measurements

In addition to the striking PL shift, the emergence of interlayer excitons can also drastically affect the optoelectronic charge transport through the MoSe₂/WS₂ interface. We have measured photocurrent in another BN-encapsulated MoSe₂/WS₂ heterobilayer (Device 2), which has source and drain contacts with a SiO₂/Si back gate (Fig. 3a). We first characterize the device by measuring the interlayer current with no optical illumination as a function of source-drain voltage ($V_{sd}$) and gate voltage ($V_g$) at room temperature. $V_{sd}$ is applied to the WS₂ flake and current is measured from MoSe₂ (Fig. 3a). The interlayer current is small at negative $V_g$ and becomes increasingly large at positive $V_g$ (Fig. 3b). This indicates $n$-type transport mediated by electrons in the conduction bands. At constant positive $V_g$, the current is nearly zero at $V_{sd} < 0$, but increases dramatically with an exponential turn-on at $V_{sd} > 0$; such rectifying behavior is consistent with our band scheme in Fig. 1b, where the MoSe₂ CBM is lower than the WS₂ CBM. This further supports the intrinsic type-I alignment (see more discussions in Supplementary Information, Section 5.1).

Afterward we measure the interlayer photocurrent ($I_{pc}$) under the excitation of an ultrafast laser. We tune the laser photon energy to be $\hbar\omega = 1.49$ eV ($\lambda = 830$ nm), which is close to the MoSe₂ exciton resonant energy at room temperature (see Methods). Figure 3c displays a photocurrent color map at varying $V_{sd}$ and $V_g$. The photocurrent

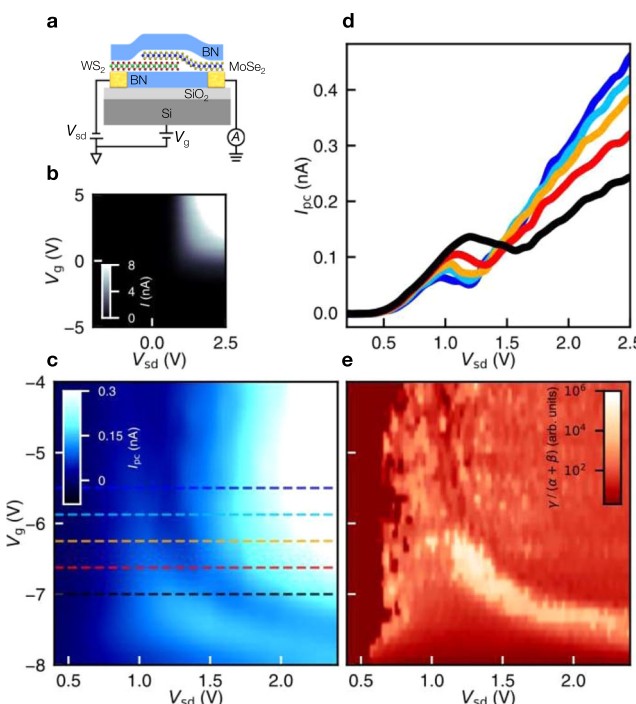

**Fig. 3 | Photocurrent of MoSe₂/WS₂ heterobilayer. a** The schematic of a MoSe₂/WS₂ heterobilayer device with silicon back gate (Device 2). **b** Grayscale map of interlayer current $I$ as a function of the source-drain voltage ($V_{sd}$) and gate voltage ($V_g$) with no optical excitation. **c** Color map of interlayer photocurrent $I_{pc}$ vs. $V_{sd}$ and $V_g$ and (**d**) Cross-cut $I_{pc}$-$V_{sd}$ profiles extracted from panel c at specific $V_g$ values, denoted by dashed lines with corresponding colors. **e** Color map of the log of the relative rates of multi-particle decay (Auger processes) to single electron-hole pair decay $\gamma/(\alpha+\beta)$ as a function of $V_{sd}$ and $V_g$ over the same range as in (**c**).

occurs predominantly in forward bias ($V_{sd} > 0$ V applied to WS₂), consistent with the type I band alignment shown in Fig. 1b. Notably, the photocurrent map exhibits a narrow, curved region where $I_{pc}$ drops with increasing $V_{sd}$. Such an anomalous suppression of interlayer photocurrent can be seen clearly from the $I_{pc}$-$V_{sd}$ line traces at different $V_g$ (Fig. 3d). As $V_{sd}$ increases, the interlayer photocurrent first increases, then drops in the range of $V_{sd} = 1$–2 V, and afterward increases again. Similar non-monotonic $I_{pc}$-$V_{sd}$ characteristics are found for a wide range of $V_g$ from −3.0 to −7.5 V, where the suppression region shifts from lower to higher $V_{sd}$ values (Fig. 3c). This shift of suppression region with $V_g$ is likely due to the change of contact resistance when the carrier density is modulated by the global silicon gate. A higher contact resistance means a higher $V_{sd}$ is required to drive an equivalent interlayer voltage drop across the heterobilayer. By imaging the spatial photocurrent response, we confirm that this photocurrent suppression only occurs in the heterobilayer region (Supplementary Fig. S7).

We have extracted the interlayer electric field ($E$) from $V_{sd}$ by modeling the device as a p-n junction and fitting its charge transport data (see Supplementary Information, Section 2.2). The anomalous suppression of photocurrent starts near $E \sim 0.23$ V/nm, comparable to the critical field ($E \sim 0.16$ V/nm) of interlayer exciton formation observed in the PL map (right axis in Fig. 2c). This suggests that the photocurrent suppression is induced by the interlayer exciton formation.

To clarify the origin of the photocurrent suppression, we have investigated its dependence on the excitation laser power $P$. Figure 4a shows the photocurrent $I_{pc}$ at varying $V_{sd}$ (top axis) and electric field $E$ (bottom axis) under increasing laser power at $V_g = -6.5$ V. At low laser power, $I_{pc}$ increases monotonically with increasing $V_{sd}$. At high laser power, however, $I_{pc}$ drops in the range of $V_{sd} = 1.0$–1.5 V. As the laser power increases, the photocurrent

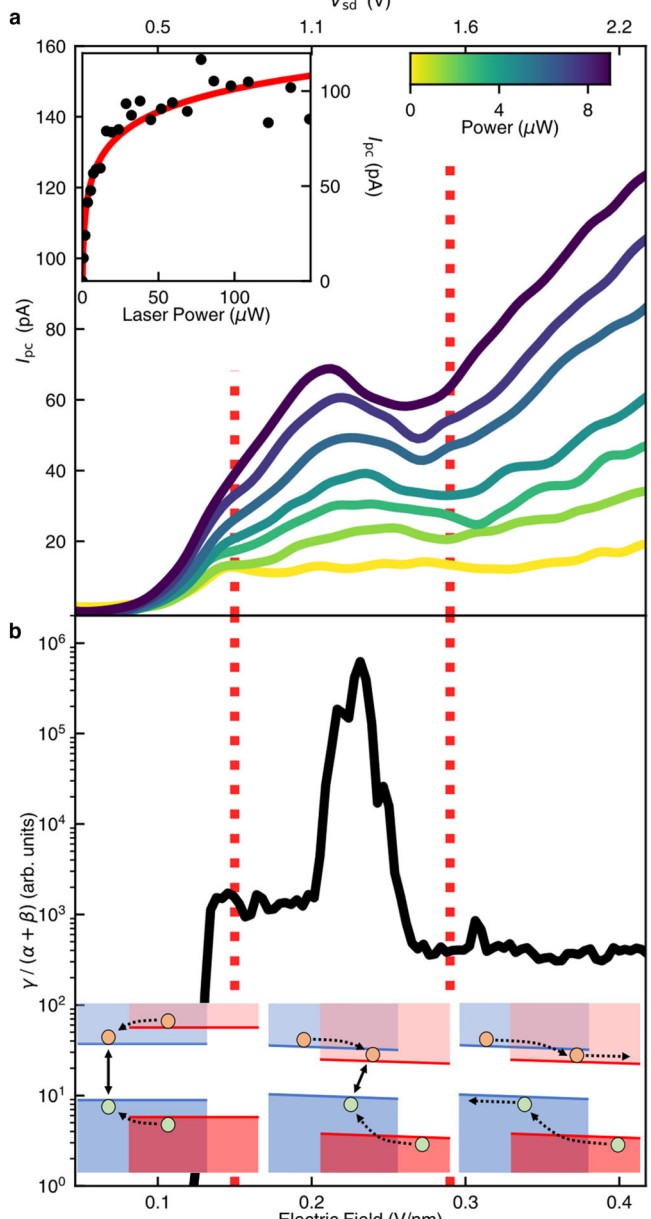

**Fig. 4 | Photocurrent signature of type-I to type-II band alignment transition in MoSe₂/WS₂ heterobilayer. a** Photocurrent ($I_{pc}$) as a function of source-drain voltage ($V_{sd}$, top axis) and interlayer electric field (bottom axis) under various excitation laser powers. The color of each curve represents the excitation power, as indicated by the color scale bar. The inset shows the photocurrent with increasing laser power at $V_{sd} = 1.15$ V ($E = 0.24$ V/nm). The red line is a fit based on the model described in the text. The gate voltage is $V_g = -6.5$ V in all measurements. **b** The best-fit $\gamma/(\alpha+\beta)$ value as a function of electric field. The vertical dashed lines approximately define three regions that correspond to the three scenarios depicted by the lower insets, which, from left to right, illustrate the absence, formation and dissociation of interlayer excitons at increasing field. The measurements were conducted at 830-nm excitation wavelength at room temperature.

suppression becomes more severe. By examining the power dependence of $I_{pc}$ at varying $V_{sd}$ and $V_g$, we find that the photocurrent generally increases sublinearly with the laser power (e.g., see the inset of Fig. 4a) and, remarkably, the degree of sublinearity is closely related to the photocurrent suppression.

The photocurrent dynamics under pulsed excitation can be captured by using a simple model. Each laser pulse instantaneously generates $N_0$ electron-hole pairs, which drops to zero population before

the next pulse arrives. The photocurrent is obtained as

$$I_{PC} = ef\alpha \int_0^\infty N(t)\mathrm{d}t \qquad (1)$$

Here $e$ is the elementary charge; $f$ is the pulse repetition rate; $\alpha$ is the carrier extraction rate; $N(t)$ is the time-dependent population of electron-hole pairs after the pulsed excitation. $N(t)$ decays according to the equation

$$dN/dt = -(\alpha+\beta)N - \gamma N^2. \qquad (2)$$

Here $\beta$ is the recombination rate of the electron-hole pairs. The $-(\alpha+\beta)N$ term describes the decrease of carrier population due to the extraction and first-order recombination of photocarriers. $\gamma$ is the decay rate due to exciton-exciton annihilation or Auger process. When exciton density $N$ increases at increasing laser power, the nonlinear decay term $(-\gamma N^2)$ becomes more important, leading to faster exciton decay and photocurrent suppression.

Solving Eq. (2) gives

$$N(t) = \frac{N_0 e^{-(\alpha+\beta)t}}{1 + \frac{\gamma N_0}{\alpha+\beta}\left(1 - e^{-(\alpha+\beta)t}\right)} \qquad (3)$$

which, when combined with Eq. (1), yields an analytical expression for the photocurrent under pulsed excitation:

$$I_{PC} = \frac{ef(\alpha+\beta)}{\gamma}\ln\left(\frac{\gamma N_0}{\alpha+\beta} + 1\right) \qquad (4)$$

By assuming that $N_0$ scales linearly with the laser power ($P$), we can use Eq. (4) to fit the $I_{pc}$–$P$ data. The fitting is excellent for a broad range of $V_{sd}$ and $V_g$ values (e.g., see the line in the inset of Fig. 4a). From the fitting, we extract the relative rate $\gamma/(\alpha+\beta)$ (up to a proportionality constant) between multi-particle decay (Auger-like processes) and single electron-hole pair decay at different $V_{sd}$ and $V_g$ (Fig. 3e).

Figure 4b displays the best-fit $\gamma/(\alpha+\beta)$ value as a function of electric field at $V_g = -6.5$ V. We observe a striking peak of $\gamma/(\alpha+\beta)$ in the range $V_{sd} = 1$–1.5 V, which coincides with the $E$-field range where the photocurrent is suppressed in Fig. 4a. To consolidate this observation, we extract the $\gamma/(\alpha+\beta)$ value at varying $V_{sd}$ and $V_g$ (Fig. 3e) and compare it with the photocurrent map (Fig. 3c). The $\gamma/(\alpha+\beta)$ enhancement is found to coincide well with the photocurrent suppression. As the carrier extraction rate $\alpha$ and the Auger-like decay rate $\gamma$ typically evolve smoothly with electric field, the sharp peak of $\gamma/(\alpha+\beta)$ implies a sudden decrease of the electron-hole recombination rate $\beta$. This is consistent with the formation of interlayer excitons in a type-I to type-II transition because the interlayer excitons have much smaller recombination rate (longer lifetime) than the intralayer excitons due to the spatial separation of electrons and holes.

The insets of Fig. 4b illustrate how the interlayer exciton formation may account for the observed photocurrent behavior. At weak interlayer electric field (left inset), the heterobilayer has the type-I band alignment and hence exhibits rectifying behavior, in which the photocurrent increases monotonically with increasing interlayer field. When the electric field reaches a critical value $E_c$ - 0.15 V/nm (comparable to $E_c$ - 0.16 V/nm in the PL results in Fig. 2c), the heterobilayer transitions from type-I to type-II band alignment, enabling the formation of interlayer excitons (middle inset). The interlayer exciton formation traps the carriers at the interface, simultaneously suppressing the photocurrent and reducing the photocarrier decay rate $\beta$ (i.e. boosting $\gamma/(\alpha+\beta)$). When the increasing electric field becomes strong enough to dissociate the interlayer excitons (right inset), the exciton effect subsides and the photocurrent resumes its normal increasing trend with increasing $V_{sd}$.

Besides Devices 1 and 2 presented above, we have also measured Devices 3 to reproduce the major PL results and Device 4 to reproduce both the major PL and photocurrent results (see Supplementary Information, Sections 1, 6 and 7).

In summary, we demonstrate controlled on/off switching of interlayer excitons in $MoSe_2/WS_2$ heterobilayers through a type-I to type-II transition, which substantially influences the optical properties and photocurrent behavior. This phenomenon stems from the closely aligned conduction band minima with field-tunable offset between monolayer $MoSe_2$ and $WS_2$, and it is not expected to occur in other TMD heterobilayers with large band offsets[47]. Our findings establish $MoSe_2/WS_2$ heterobilayers as a highly adaptable platform for excitonic research and applications. For instance, one may harness this effect for switching a hypothetical interlayer excitonic Bose-Einstein condensate, tuning exciton potential depth, and realizing depth-adjustable exciton traps within van der Waals heterostructure materials. The tuning mechanism complements other 'live' tunable parameters, such as strain, stress, and twist angles, and achieves 'in-situ' control of band-engineered exciton behaviors. This integration promises a new level of precision and adaptability in manipulating excitonic properties in these advanced materials.

## Methods

### Device fabrication

All $MoSe_2/WS_2$ heterobilayer devices are fabricated by applying a polycarbonate-based dry-transfer technique to stack different 2D crystals together. The substrates are silicon wafers with 300-nm-thick oxide layer. For the dual-gate Devices 1, 3, 4, we use a polycarbonate stamp to sequentially pick up a thin graphite flake (serving as the top-gate electrode), a thin BN flake (as the top-gate dielectric), monolayer $MoSe_2$, monolayer $WS_2$, a second thin graphite flake (as the contact electrode), another thin BN flake (as the bottom gate dielectric), and a third thin graphite flake (as the bottom-gate electrode). During the stacking process, we align the sharp edges of the $MoSe_2$ and $WS_2$ crystals so that the twist angles between them are expected to be close to 0° or 60°. Afterward, we deposit the stack of materials onto the $Si/SiO_2$ substrate. Finally, we use the standard electron-beam lithography to deposit the gold contacts (70-nm thickness) onto the devices.

For single-gate Device 2 used in the photocurrent experiment, we first use electron-beam lithography to deposit the two gold contacts (as source and drain electrodes) on a $Si/SiO_2$ substrate. Afterward, we use a polycarbonate stamp to transfer a BN flake to cover the area between the two electrodes. Upon this surface with pre-patterned electrodes, we transfer a $MoSe_2/WS_2$ heterobilayer stack by using a large thin BN flake to sequentially pick up monolayer $MoSe_2$ and monolayer $WS_2$. We align the sample position so that the $WS_2$ layer contacts one electrode and the $MoSe_2$ layer contacts the other electrode. This allows us to apply a bias voltage between the two layers.

### Photoluminescence experiments

The photoluminescence (PL) experiments are performed in a closed-cycle cryostat (Montana), where the sample temperature is estimated to be $T$ - 6 K. The excitation light source is a 532-nm continuous-wave laser (Torus 532, Laser Quantum). The laser is focused onto the sample with a spot diameter of 1-2 μm by an objective lens (numerical aperture 0.6). The incident laser power is $P$ - 3 μW. The PL is collected by the same objective and analyzed by a spectrometer (HRS-500-MS, Princeton Instruments) equipped with a charge-coupled-device (CCD) camera. Two Keithley K2400 source meters are used to independently control the top and bottom gate voltages.

### Photocurrent experiments

Photocurrent experiments are performed in vacuum in a customized Janis Research ST-3T-2 optical cryostat. Device 2 is measured at room

temperature; Device 4 is measured at T = 50 K. The light source is an ultrafast Coherent Mira laser that generates pulses with 150-fs duration and 75-MHz repetition rate. The laser wavelength is tuned to either 790 nm or 830 nm, close to the optical band gap of monolayer $MoSe_2$ as well as the energy of the interlayer exciton. The laser is focused onto the sample with a spot diameter of ~2 μm using a Thorlabs gradient-index (GRIN) lens. The laser position on the sample is controlled by a Thorlabs galvanometer. The galvo position, $V_{sd}$, and $V_g$ are controlled by two data acquisition cards (DAQs) from National Instruments. We measure the interlayer current with a pre-amplifier (DL Instruments 1211). The optically induced current is extracted from the total current by using a lock-in amplifier (Stanford Research) and optical chopper.

## Data availability

The data generated in this study have been deposited into https://github.com/qmolabucr/EField-Tunable-MoSe2WS2. This repository includes all the relevant data and the python scripts that are used to generate the figures such that the results can be fully replicated.

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

## Acknowledgements

This work was supported by the Army Research Office Electronics Division Award no. W911NF2110260 (N.M.G., V.A., and J.K.-M.), the Presidential Early Career Award for Scientists and Engineers (PECASE) through the Air Force Office of Scientific Research (award no. FA9550-20-1-0097; N.M.G. and T.B.A.), through support from the National Science Foundation Division of Materials Research CAREER Award (no. 1651247; N.M.G. and J.K.-M.), and through the United States Department of the Navy Historically Black Colleges, Universities and Minority Serving Institutions (HBCU/MI) award no. N00014-19-1-2574 (N.M.G. and T.B.A.). C.H.L. acknowledges support from the National Science Foundation (NSF) Division of Materials Research CAREER Award No. 1945660 and from the American Chemical Society Petroleum Research Fund No. 61640-ND6. K.W. and T.T. acknowledge support from the JSPS KAKENHI (Grant Numbers 19H05790, 20H00354 and 21H05233).

## Author contributions

A.S., E.L. and J.K.-M. fabricated the devices. A.S. and E.L. performed the photoluminescence and reflectance contrast experiments. J.K.-M. and F.F. performed the photocurrent experiments. T. A. and N.M.G. designed the photocurrent experiments, T.A. and J.K.-M. obtained initial photocurrent data that stimulated further study. V.A. provided theoretical support to the interpretation of the experimental data. T.T. and K.W. provided boron nitride crystals. C.H.L. and N.M.G. supervised the project, while C.H.L., A.S., J.K.-M. wrote the manuscript with input from all other authors.

## Competing interests

The authors declare no competing interests.
