## [Peer Review File · Nature Communications]

Electric-field tunable Type-I to Type-II band alignment transition in MoSe₂/WS₂ heterobilayersREVIEWER COMMENTS

Reviewer #1 (Remarks to the Author):

The authors report an interesting experimental demonstration of tunable type-I to type-II transition based on a doubly-gated transition-metal dichalcogenide heterobilayer system, here composed of MoSe₂/WS₂ due to the favorable band-gaps alignment configuration. In other words, assuming an electric-field tunable switching from straddling to staggered gap (i.e. type-I to type-II) heterojunction, this work discusses how the vertically applied fields affect the alignment configuration. By probing optically and electrically the relevant excitonic modes respectively conductivity of the optically-driven and electronically gated heterostructure system, the band alignment change is reported by monitoring both optical and photocurrent signatures in the MoSe₂/WS₂ bilayer.

Interlayer exciton photoluminescence was turned on and off as a function of the applied field strength and suppression of carrier separation across the interface evidenced by photocurrent features. The methodology is described and applied in a sound fashion and the observations analyzed with the help of their modeling work. Additional considerations incl. field-dependent reflection contrast data, device property mappings, etc., in support of the study are delivered as supplementary materials. The interesting results and findings appear to be sufficiently discussed and explained in the context of their 2D heterostructure research. Yet, the authors could place their own efforts into the context of ongoing research with a bit more attention to the existing literature on type-I and type-II transitions, and use less sensational expressions to wrap-up their research outcome for the (wider) semiconductor community.

As part of manuscript revisions, the authors' should address the following remarks before publication of their work in this journal:

1. The expression "such a material has never been realized" seems not appropriate for the abstract. Rather suitable would be: "Because it is desirable [...], such an outstanding heterojunction configuration is experimentally demonstrated in this work". Or "[...], a so-far unique heterostructure example is experimentally demonstrated in this work".
2. Further title refinement possible, such as towards "Electrically-tunable type-I to type-II transition in ...heterobilayers with on-/off-switchable interlayer exciton luminescence"
3. The main text states "has not yet been demonstrated". The authors had probably in mind to say "has been proposed but – to the best of the authors' knowledge – not demonstrated".
4. The introduction is well composed to give the wider audience an idea of the aim and the addressed material system to solve this engineering task. However, the last part or paragraph lacks a necessary comparison to, as well as discussion of, other concepts in the literature proposed or demonstrated to observe electrically- and non-electrically-tunable type-I and type-II transition. A quick survey in the web gives a number of relevant articles that should be appropriately taken into account. Without sorting and prioritizing any, a few of them are listed here:
 - 4a. "Vertical strain and electric field tunable electronic properties of type-II band alignment C₂N/InSe van der Waals heterostructure", Pham et al. <https://doi.org/10.1016/j.cplett.2018.12.027>. The work's abstract states for instance "The type-II band alignment can be switched to type-I one and an indirect to direct band gap transition can be achieved by applying the electric field or vertical strain."
 - 4b. "Tunable type-II band alignment and electronic structure of C₃N₄/MoSi₂N₄ heterostructure: Interlayer coupling and electric field", Nguyen et al. <https://journals.aps.org/prb/abstract/10.1103/PhysRevB.105.045303>, stating "Remarkably, the electronic structure and the band alignment types can be flexibly tuned between type-I and type-II by applying an external electric field[...]"
 - 4c. One should also give examples, how tunability can be else realized, e.g. in the abstract of <https://doi.org/10.1016/j.cplett.2018.12.027>, it is further stated that "[...]can be flexibly tuned

between type-I and type-II by applying an external electric field, by changing the interlayer distance and by applying the in-plane strain".

4d. Examples are also "Inverted Core/Shell Nanocrystals Continuously Tunable between Type-I and Type-II Localization Regimes" Balet et al., <https://pubs.acs.org/doi/abs/10.1021/nl049146c>

4e. or "Tunable type-I/type-II transition in g-C₃N₄/graphyne heterostructure by BN-doping: A promising photocatalyst" Yun et al., <https://doi.org/10.1016/j.solmat.2020.110516>

4f. In view of "twistronics", also "Rotation Tunable Type-I/Type-II Band Alignment and Photocatalytic Performance of g-C₃N₄/InSe van der Waals Heterostructure", Chang et al., <https://doi.org/10.1002/pssr.202100171> appears relevant for recently proposed manually tunable twist devices (by some groups of the 2D community)

4g. Other concepts utilize different material platforms to discuss transitions of this type such as in "Tunable Type I and II heterojunction of CoOx nanoparticles confined in g-C₃N₄ nanotubes for photocatalytic hydrogen production", Zhu et al., <https://doi.org/10.1016/j.apcatb.2018.12.015>

5. Authors state in the first paragraph of section "Photoluminescence measurements" that (some weak lines) "[...]are not reproducible in another device (Fig. S1) and probably arise from the influence of moiré patterns in the heterostructure". A brief discussion regarding possible atomic reconstruction (for those quite small twist angles close to 0°) as an alternative to presumed moiré patterns formed by heterostructuring the bilayer system and the impact of any such regime (moiré vs. reconstruction) for their experiment seems recommendable.

6. Why haven't the authors included a more off-symmetry twist angle in their study? It would be helpful if they highlight in the manuscript the main features and expected behaviors of these rather distinct cases when comparing near 0°/60° and off-0°/60° twists.

7. The final paragraph of the manuscript concludes with a bit of exaggeration and imprecise statements:

7a. Engineering-wise a remarkable demonstration, but "fascinating phenomenon" (last paragraph, page 6) is a bit too much for this effect, branding it "useful phenomenon" or "extraordinarily well-tuned condition" seems reasonable, though!

7b. "[...], one may realize electrical on/off switching of the Bose-Einstein condensate in [...] heterobilayers" should not be sold this way. One can accept rather a moderate claim such as "[...] may harness on this effect for the switching of an/any (interlayer) excitonic Bose-Einstein condensate [...]". First of all, condensates of such kind (or similar phase transitions based on a spontaneous symmetry break, and usually at quite low temperatures) need to be first of all convincingly and unambiguously demonstrated with 2D heterostructures. Secondly, their fragile nature let's the external field applied impact them in a yet-to-be carefully examined way, letting one believe there exist much more subtle ways the field will matter in those studies than merely switching interlayer existence and, with it, condensate formation "on" and "off".

7c. "demonstrate electrically tunable exciton lattices [...]" to be corrected to "lattice potential depths" (to be specific, no other control over the – if formed – "moiré" lattice sites or super cells is anticipated/can be deduced by the authors from this work here)

7d. For reasons not further discussed, the last sentence should be carefully revised in the sense that instead of stating (originally) "Akin to early research in optical trapping - in which atoms were controllably trapped in optical lattices - our experiments represent a keystone in our understanding of trapping charge carriers into tunable exciton traps in van der Waals heterostructures materials and give promise for ever greater control over excitons in band engineered devices.", it appropriately may read for instance (revised sentence:) "[...] our experiments represent a pathway towards electronically-tunable trapping of charge carriers into depth-adjustable exciton traps in van der Waals heterostructures materials and thereby promise a practical (device's) tuning knob next to other 'live'-tunable parameters (e.g. strain/stress and twists) for 'in-situ' control of band-engineered exciton behaviors." (again, see previous comment, about no specific control over lattice sites or moiré cell sizes)

8. Miscellaneous/Cosmetics:

8a. Should be "arb. u." wherever authors refer to arbitrary units in labels (see inhomogeneous use)

In summary, the interesting work with novel character in the experimental domain can become suitable for publication soon, provided that reviewers' remarks are carefully addressed.

Reviewer #2 (Remarks to the Author):

In this paper, J. Kistner-Morris and colleagues fabricated TMDs heterobilayers and investigated them by means of PL and PC spectroscopy. They demonstrated bandgap alignment tunability (from type-I to type-II) by applying electric fields through the layers. They then present and discuss a resultant on/off switching behavior of the interlayer excitons. Overall, the paper is clear and scientifically sound. The topic of interlayer excitons in TMDs materials is of interest to the readers of Nature Communications and the on/off modulation has potential relevance for technological applications. Nevertheless, some amendments are necessary to the text.

Please, my comments are listed detailed below:

- 1.1 – Can the authors comment if such a switching effect happens in other TMD heterobilayers?
- 1.2 – Device design #2 used SiO₂/Si as back-gate instead of graphite gates, what is the reason? and how did the author avoid photodoping effects or consider them in the analysis?
- 1.3 – The authors should test the I_{pc} measurements under illumination on graphite-gated samples to eliminate any contribution from SiO₂/Si surfaces.
- 1.4 – The authors speculate interlayer exciton in this system (Dev02), but PL was shown. Can the authors provide such measurements?
- 1.5 – Can the authors add the SiO₂ thickness used in the fabrication? The V_g values applied on Device #2 are quite low (down to -8V), thus, depending on the dielectric thickness used, the electric field is very low too. The field level created in device #1,3 is much larger than that in device #2, considering the back-gate. In Device#2 the authors claim a change in band alignment due to V_{sd}, what is the electric field value in this case? Is the total E_{field} comparable to the V_g field in #1,3? The authors should clarify this.
- 1.6 – The authors presented in the SI some characterizations which were not mentioned in the main text. Please, add a proper indication. Moreover, the authors mentioned details of measurements in the SI which were not referenced in the main text "i.e., V_{sd} was applied to the WS₂ flake and current was measured from MoSe₂". This was not mentioned in the main text. Please move it to the main text since this helps the understanding of the device.

Reviewer #3 (Remarks to the Author):

The manuscript presents research work that reveals the switch between the type-I and type-II band alignments in MoSe₂/WS₂ heterobilayers. By applying a vertical electric field, luminescence and photocurrent measurements show that the MoSe₂/WS₂ bilayer can be switched from a type-I band alignment to a type-II band alignment. The authors further developed a phenomenological model to show that the highly nonlinear photocurrent, according to the source-drain electric field, is due to the transition of band alignments and substantial changes in exciton lifetime. The work utilizes the unique layer structures of van der Waals (vdW) materials to achieve the electric-field tunable band alignment, which is challenging in non-vdW materials. This result can be useful for new optoelectronic device applications.

Regarding the current version of the manuscript, I have a few questions:

1. I am not sure about the type-I band alignment in intrinsic MoSe₂/WS₂ heterobilayers. I noticed that there were a few calculations (e.g., Nano Lett. 2016, 16, 7, 4087–4093) showing that it is a type-II band alignment. This manuscript confirms that the CBM is from MoSe₂, which is not enough to confirm the overall type-I band alignment. Moreover, the electron-hole binding energies

for intralayer and interlayer excitons are different, and it is questionable to extrapolate the band alignment from the exciton energies in Figure 2c. It is better to employ theoretical calculations of the band structure and excitons to support the claims of this work.

2. The field dependence of the lifetime of excitons shown in Figure 4d is interesting. Can the authors convert the V_{sd} to the value of an electric field? I hope to know if it agrees with the crucial field (0.16 V/nm) observed in luminescence measurements.

3. The explanation of the laser power dependence is not clear to me. It seems that higher power introduces more significant nonlinear curves in Figure 4a. Does it affect the lifetime of intralayer or interlayer excitons?

RESPONSE TO REVIEWERS' COMMENTS

Reviewer #1 (Remarks to the Author):

Comments: The authors report an interesting experimental demonstration of tunable type-I to type-II transition based on a doubly-gated transition-metal dichalcogenide heterobilayer system, here composed of MoSe₂/WS₂ due to the favorable band-gaps alignment configuration. In other words, assuming an electric-field tunable switching from straddling to staggered gap (i.e. type-I to type-II) heterojunction, this work discusses how the vertically applied fields affect the alignment configuration. By probing optically and electrically the relevant excitonic modes respectively conductivity of the optically-driven and electronically gated heterostructure system, the band alignment change is reported by monitoring both optical and photocurrent signatures in the MoSe₂/WS₂ bilayer.

Interlayer exciton photoluminescence was turned on and off as a function of the applied field strength and suppression of carrier separation across the interface evidenced by photocurrent features. The methodology is described and applied in a sound fashion and the observations analyzed with the help of their modeling work. Additional considerations incl. field-dependent reflection contrast data, device property mappings, etc., in support of the study are delivered as supplementary materials. The interesting results and findings appear to be sufficiently discussed and explained in the context of their 2D heterostructure research.

Response: We thank the reviewer for reading our manuscript and the positive remarks. Below we will give our point-to-point reply to the comments. We have revised our manuscript and also added new contents and re-organized the Supplementary Information. In the revised manuscript, we have highlighted the relevant changes in red color for the convenience of reviewers.

Comments: Yet, the authors could place their own efforts into the context of ongoing research with a bit more attention to the existing literature on type-I and type-II transitions, and use less sensational expressions to wrap-up their research outcome for the (wider) semiconductor community.

Response: We thank the reviewer for this constructive comment. We have revised our introduction to include broader research context and also tuned down the sensational expression in the text.

Comments: As part of manuscript revisions, the authors should address the following remarks before publication of their work in this journal:

The expression “such a material has never been realized” seems not appropriate for the abstract. Rather suitable would be: “Because it is desirable [...], such an outstanding heterojunction configuration is experimentally demonstrated in this work”. Or “[...], a so-far unique heterostructure example is experimentally demonstrated in this work”.

Response: We thank the reviewer for the suggestion. We have revised the expression in the abstract into “*The distinct characteristics and applications associated with each alignment make it highly desirable to switch between them within a single material. Here we demonstrate an electrically tunable transition between type-I and type-II band alignments in MoSe₂/WS₂ heterobilayers by investigating their luminescence and photocurrent characteristics.*”

Comment: Further title refinement possible, such as towards “Electrically-tunable type-I to type-II transition in ...heterobilayers with on-/off-switchable interlayer exciton luminescence”

Response: We have changed the title into “*Electrically tunable Type-I to Type-II band alignment transition in MoSe₂/WS₂ heterobilayers*”.

Comment: The main text states “has not yet been demonstrated”. The authors had probably in mind to say “has been proposed but – to the best of the authors’ knowledge – not demonstrated”.

Response: We thank the reviewer for suggesting this refinement. The containing paragraph has been reworked and the final sentence now reads, “*A more efficient alternative envisions the capability to tune the band alignment types through straightforward electrical means in a single device, though, to the best of our knowledge, such a demonstration has not been realized.*”

Comment: The introduction is well composed to give the wider audience an idea of the aim and the addressed material system to solve this engineering task. However, the last part or paragraph lacks a necessary comparison to, as well as discussion of, other concepts in the literature proposed or demonstrated to observe electrically- and non-electrically-tunable type-I and type-II transition. A quick survey the web gives a number of relevant articles that should be appropriately taken into account. Without sorting and prioritizing any, a few of them are listed here:

4a. “Vertical strain and electric field tunable electronic properties of type-II band alignment C₂N/InSe van der Waals heterostructure”, Pham et al. <https://doi.org/10.1016/j.cplett.2018.12.027>. The work’s abstract states for instance “The type-II band alignment can be switched to type-I one and an indirect to direct band gap transition can be achieved by applying the electric field or vertical strain.”

4b. “Tunable type-II band alignment and electronic structure of C₃N₄/MoSi₂N₄ heterostructure: Interlayer coupling and electric field”, Nguyen et al. <https://journals.aps.org/prb/abstract/10.1103/PhysRevB.105.045303>, stating “Remarkably, the electronic structure and the band alignment types can be flexibly tuned between type-I and type-II by applying an external electric field[...].”

4c. One should also give examples, how tunability can be else realized, e.g. in the abstract of <https://doi.org/10.1016/j.cplett.2018.12.027>, it is further stated that “[...]can be flexibly tuned between type-I and type-II by applying an external electric field, by changing the interlayer distance and by applying the in-plane strain”.

4d. Examples are also “Inverted Core/Shell Nanocrystals Continuously Tunable between Type-I and Type-II Localization Regimes” Balet et al., <https://pubs.acs.org/doi/abs/10.1021/nl049146c>

4e. or “Tunable type-I/type-II transition in g-C₃N₄/graphyne heterostructure by BN-doping: A promising photocatalyst” Yun et al., <https://doi.org/10.1016/j.solmat.2020.110516>

4f. In view of “twistronics”, also “Rotation Tunable Type-I/Type-II Band Alignment and Photocatalytic Performance of g-C₃N₄/InSe van der Waals Heterostructure”, Chang et al.,

<https://doi.org/10.1002/pssr.202100171> appears relevant for recently proposed manually tunable twist devices (by some groups of the 2D community)

4g. Other concepts utilize different material platforms to discuss transitions of this type such as in “Tunable Type I and II heterojunction of CoOx nanoparticles confined in g-C3N4 nanotubes for photocatalytic hydrogen production”, Zhu et al., <https://doi.org/10.1016/j.apcatb.2018.12.015>

Response: We thank the reviewer for the detailed survey of these relevant articles. We have revised the introduction to include these researches of band alignment engineering as follows:

(1) For papers in general materials, such as 4d, 4e, 4g, we cited them in the second paragraph on Page 2: “*While certain approaches involve material engineering through adjusting chemical compositions^{15–18} or layer thickness¹⁹, these methods necessitate multiple synthesis iterations, precise dopant control, or the use of different samples.*”

(2) For theoretical proposals related to 2D vdW heterostructures, such as 4a, 4b, 4f, we cited them in the third paragraph on Page 2: “*Indeed, recent theoretical propositions have delved into the possibility of tuning band alignments in 2D vdW heterostructures through various means, including electric field^{38,39}, strain³⁸, interlayer spacing³⁸, and twist angle⁴⁰.*”

We hope that these revisions have properly addressed the ongoing research on band alignment engineering.

Comment: Authors state in the first paragraph of section “Photoluminescence measurements” that (some weak lines) “[...]are not reproducible in another device (Fig. S1) and probably arise from the influence of moiré patterns in the heterostructure”. A brief discussion regarding possible atomic reconstruction (for those quite small twist angles close to 0°) as an alternative to presumed moiré patterns formed by heterostructuring the bilayer system and the impact of any such regime (moiré vs. reconstruction) for their experiment seems recommendable.

Response: We thank the reviewer for recommending the inclusion of this alternate scenario. In the revised manuscript, we briefly mention these effects, including also the sample inhomogeneities: “*We note that the weak lines at higher energies than A_{M_0} may arise from moiré effect or sample inhomogeneities, since they are not reproducible in other devices (Fig. S8, S9). Atomic reconstruction is unlikely to occur in this system due to the small moiré wavelength (~8 nm)*” It is difficult to exactly pinpoint the reason at this point based on our experimental data.

We believe that that atomic reconstruction is not likely to take place in this specific heterobilayer, as monolayer MoSe₂ and WS₂ have 4% lattice mismatch (lattice constants are 3.32 Å for MoSe₂ and 3.18 Å for WS₂). This lattice mismatch creates a rigid moiré pattern with a moiré wavelength of ~8 nm even for twist angles exactly at 0° or 60°. Such small moiré wavelength places the heterobilayer out of the regime where atomic reconstruction starts to dominate (typically with moiré wavelengths above tens of nanometers [*Nat Mater* 20, 480–487 (2021)]). Atomic reconstruction is so far only observed in twisted homobilayers [e.g., *Nat Mater* 18, 448–453 (2019), *Nat Mater* 20, 480–487 (2021)], or heterobilayers where the constituent layers have very small lattice mismatch, such as MoSe₂/WSe₂ or MoS₂/WS₂ [e.g., *Nat Nanotech* 15, 592–597 (2020), *ACS Nano* 2020, 14, 4, 4550–4558]. For similar heterobilayers such as WSe₂/WS₂ where there is ~4% lattice mismatch, moiré superlattice is rigidly formed [e.g., *Nature* 579, 353–358 (2020), *Phys Rev Lett* 127, 037402 (2021)].

Comment: Why haven't the authors included a more off-symmetry twist angle in their study? It would be helpful if they highlight in the manuscript the main features and expected behaviors of these rather distinct cases when comparing near $0^\circ/60^\circ$ and Off- $0^\circ/60^\circ$ twists.

Response: We thank the reviewer for the suggestion. We expect similar type-I to type-II band alignment transition in non- $0^\circ/60^\circ$ heterobilayers. However, these twisted heterobilayers exhibit a large momentum mismatch between the K valleys of the two layers due to their relative rotation. The interlayer momentum mismatch strongly quenches the photoluminescence of interlayer excitons [*Nature* 567, 66–70 (2019)]. Therefore, it is important to use $0^\circ/60^\circ$ heterobilayers to review the optical signatures of the band alignment transition.

We fabricated and measured another dual-graphite-gate MoSe₂/WS₂ heterobilayer device (Device 5) with a large twist angle of about 30 degrees. Figure R1a displays its charge-density-dependent PL map. The map shows the dominant PL features of the intralayer exciton (A_{Mo}) and trions (A_{Mo}^- , A_{Mo}^+) from the MoSe₂ layer, consistent with the PL results in the main manuscript. Figure R1b displays the electric-field-dependent PL map. We still observe the interlayer exciton (IX) at large electric field, but the PL intensity is very weak. To highlight the weak IX feature, we intentionally tune down the color scale and saturate the intralayer excitonic features. The results of Device 5 show that the type-I to type-II transition still exists in MoSe₂/WS₂ heterobilayer with a large twist angle, but the interlayer PL is suppressed due to the interlayer momentum mismatch.

In the revised manuscript (Page 3), we briefly comment this issue by writing “*The heterobilayers for PL measurements have twist angles of either $\sim 0^\circ$ or $\sim 60^\circ$. Deviation from these angles will suppress the interlayer emission due to electron-hole momentum mismatch.*”

Figure R1 | Gate-dependent photoluminescence maps of MoSe₂/WS₂ heterobilayer Device 5. **a**, The charge-density-dependent PL map. Equal voltages $V_{bg} = V_{tg}$ are applied to the bottom and top gates. The charge density is proportional to the gate voltages. **b**, The electric-field-dependent PL map. Opposite voltages $V_{bg} = -V_{tg} - 4$ volts are applied to the bottom and top gates to induce an interlayer electric field (left axis). The color scale of this map is intentionally adjusted to reveal the weak emissions from the interlayer excitons. All measurements were performed with 532-nm laser excitation (incident power $\approx 3 \mu\text{W}$) at sample temperature $T \approx 6$ K, the same as the conditions for Device 1 in Figure 2 of the main paper.

Comment: The final paragraph of the manuscript concludes with a bit of exaggeration and imprecise statements:

7a. Engineering-wise a remarkable demonstration, but “fascinating phenomenon” (last paragraph, page 6) is a bit too much for this effect, branding it “useful phenomenon” or “extraordinarily well-tuned condition” seems reasonable, though!

7b. “[...], one may realize electrical on/off switching of the Bose-Einstein condensate in [...] heterobilayers” should not be sold this way. One can accept rather a moderate claim such as “[...] may harness on this effect for the switching of an/any (interlayer) excitonic Bose-Einstein condensate [...]”. First of all, condensates of such kind (or similar phase transitions based on a spontaneous symmetry break, and usually at quite low temperatures) need to be first of all convincingly and unambiguously demonstrated with 2D heterostructures. Secondly, their fragile nature let’s the external field applied impact them in a yet-to-be carefully examined way, letting one believe there exist much more subtle ways the field will matter in those studies than merely switching interlayer existence and, with it, condensate formation “on” and “off”.

7c. “demonstrate electrically tunable exciton lattices [...]” to be corrected to “lattice potential depths” (to be specific, no other control over the – if formed – “moiré” lattice sites or super cells is anticipated/can be deduced by the authors from this work here)

7d. For reasons not further discussed, the last sentence should be carefully revised in the sense that instead of stating (originally) “Akin to early research in optical trapping - in which atoms were controllably trapped in optical lattices - our experiments represent a keystone in our understanding of trapping charge carriers into tunable exciton traps in van der Waals heterostructures materials and give promise for ever greater control over excitons in band engineered devices.”, it appropriately may read for instance (revised sentence:) “[...] our experiments represent a pathway towards electronically-tunable trapping of charge carriers into depth-adjustable exciton traps in van der Waals heterostructures materials and thereby promise a practical (device’s) tuning knob next to other ‘live’-tunable parameters (e.g. strain/stress and twists) for ‘in-situ’ control of band-engineered exciton behaviors.” (again, see previous comment, about no specific control over lattice sites or moiré cell sizes)

Response: We thank the reviewer for these thoughtful comments. We have revised the conclusion in the manuscript (highlighted in red font in Page 7). We hope the new ending is now an appropriate conclusion of our work.

Comment:

Miscellaneous/Cosmetics:

8a. Should be "arb. u." wherever authors refer to arbitrary units in labels (see inhomogeneous use)

Response: We thank the reviewer for pointing this out. The figure labels have been corrected to be consistent throughout.

Comment: In summary, the interesting work with novel character in the experimental domain can become suitable for publication soon, provided that reviewers’ remarks are carefully addressed.

Response: We thank the reviewer again for the detailed comments that help us improve the manuscript.

Reviewer #2 (Remarks to the Author):

Comment: In this paper, J. Kistner-Morris and colleagues fabricated TMDs heterobilayers and investigated them by means of PL and PC spectroscopy. They demonstrated bandgap alignment tunability (from type-I to type-II) by applying electric fields through the layers. They then present and discuss a resultant on/off switching behavior of the interlayer excitons. Overall, the paper is clear and scientifically sound. The topic of interlayer excitons in TMDs materials is of interest to the readers of Nature Communications and the on/off modulation has potential relevance for technological applications. Nevertheless, some amendments are necessary to the text.

Please, my comments are listed detailed below:

Response: We thank the reviewer for the positive comments. Below we will give our point-to-point reply to the comments. We have revised our manuscript and also added new contents and re-organized the Supplementary Information. In the revised manuscript, we have highlighted the relevant changes in red color for the convenience of reviewers.

Comment: Can the authors comment if such a switching effect happens in other TMD heterobilayers?

Response: The electric-field-tunable type-I to type-II transition is possible in MoSe₂/WS₂ heterobilayers because monolayer MoSe₂ and WS₂ have closely aligned conduction band minima. We do not expect a similar transition in other TMD heterobilayers because they have large band offsets (~400 meV or higher) [See, e.g., *Appl Phys Lett* 102, 012111 (2013)]. For instance, the largest electric field that can be applied in a dual-gate device with boron nitride (BN) dielectric is ~0.4 V/nm before dielectric breakdown. This corresponds a bilayer Stark shift of 0.4 V/nm × 0.6 nm = 240 meV, smaller than the required ~400 meV band offset.

In the conclusion of the revised manuscript, we added this sentence, “*This phenomenon stems from the closely aligned conduction band minima with field-tunable offset between monolayer MoSe₂ and WS₂, and it is not expected to occur in other TMD heterobilayers with large band offset⁴⁷.*” to highlight the uniqueness of this phenomenon.

Comment: Device design #2 used SiO₂/Si as back-gate instead of graphite gates, what is the reason? and how did the author avoid photodoping effects or consider them in the analysis?

Response: We thank the reviewer for these questions. There is a distinctive experimental tradeoff between graphite-gate devices and silicon-gate devices. Dual graphite gates allow high-precision

control of the electric field and potential at the interface, which enables good photoluminescence measurements. However, these devices often exhibit hot carrier transfer between the gates and the heterojunction, making photocurrent mapping difficult. Additionally, BN-encapsulated dual-gate devices with source-drain contacts are more prone to accidental short circuit that hinders transport measurement. Silicon-gate devices avoid both drawbacks. Hot carrier transfer between the silicon gate and heterojunction is effectively impossible and the simpler heterojunction stack produces more reliable photocurrent results.

In Section 7.2 of the revised Supplementary Information, we also present the photocurrent results of a dual-graphite-gate Device 4. The major photocurrent features are observed in this device, though somehow the data quality is not as good as that in Device 2. Therefore, we prefer showing the photocurrent data of Device 2 in the main paper.

We observe the photodoping effect in the silicon-gate devices. The photodoping effect does not change the $I-V_g$ behavior except causing a slow drift of the V_g offset over the time scale of days or weeks.

Comment: The authors should test the I_{pc} measurements under illumination on graphite-gated samples to eliminate any contribution from SiO_2/Si surfaces. The authors speculate interlayer exciton in this system (Dev02), but PL was [not] shown. Can the authors provide such measurements?

Response: We thank the reviewer for these critical comments. We have attempted to carry out photocurrent measurement on Device 1, but found a strong photoexcited leakage current between the heterobilayer and the graphite gates, which made the photocurrent measurement impossible. On the other hand, after multiple usages, Device 2 unfortunately degraded and cannot be used for PL measurement.

To address the reviewer's concern, we have fabricated a new dual-graphite-gate device (Device 4) with thin graphite contacts (the same geometry as Device 1).

Figure R2 compares the dark current and photocurrent results between Device 4 and silicon-gate Device 2. Both devices show similar results, including the suppression behavior of photocurrent. But the silicon-gate Device 2 produces higher-quality photocurrent data, with sharper photocurrent suppression. Therefore, we present the photocurrent results of Device 2 in the main manuscript.

Figure R3 shows the PL maps of Device 4. It exhibits the emergence of interlayer exciton at high positive electric field, consistent with the results of Device 1.

Overall, we are able to reproduce the major PL results and photocurrent results in Device 4, though the data are of lower quality than those in Devices 1 and 2. The results of Device 4 are presented in Section 7 of the revised Supplementary Information.

Figure R2 | Comparison between silicon-gate Device 2 and graphite-gate Device 4. a-c, Dark transport map (a) photocurrent map (b) and the I_{pc} - V_{sd} profile at $V_g = -4.2$ V (c) for the silicon-gate Device 2 with gold contacts. d-f, Similar plots for the dual-graphite-gate Device 4 with thin graphite contacts.

Figure R3 | Gate-dependent photoluminescence (PL) maps of MoSe₂/WS₂ heterobilayer Device 4. a, The charge-density-dependent PL map. Equal voltages $V_{bg} = V_{tg}$ are applied to the bottom and top gates. The charge density is proportional to the gate voltages. b, The electric-field-

dependent PL map. Opposite gate voltages $V_{bg} = -1.25V_{tg} - 11$ volts are applied to the bottom and top gates to induce an interlayer electric field. The ratio and offset between the two serve to compensate the different gating efficiencies and residual doping in the sample. An interlayer exciton (IX) feature is observed at high electric field, signifying a type-I to type-II band alignment transition in the heterobilayer. All measurements were performed with 532-nm laser excitation (incident power $\approx 3 \mu\text{W}$) at sample temperature $T \approx 6$ K, the same as the conditions for Device 1 in Figure 2 of the main manuscript.

Comment: Can the authors add the SiO_2 thickness used in the fabrication? The V_g values applied on Device #2 are quite low (down to -8V), thus, depending on the dielectric thickness used, the electric field is very low too. The field level created in device #1,3 is much larger than that in device #2, considering the back-gate. In Device#2 the authors claim a change in band alignment due to V_{sd} , what is the electric field value in this case? Is the total E-field comparable to the V_g field in #1,3? The authors should clarify this.

Response: We thank the reviewer for this important question. The SiO_2 thickness in all samples was 300 nm. There is an 10-20-nm-thick BN between the heterobilayer and the SiO_2 surface. We have added some texts in the Methods section to specify these details.

We agree with the reviewer that we should clarify the critical electric fields in different experiments and devices. We have calculated the electric fields in both PL and photocurrent experiments (see Section 2 in the revised Supplementary Information), and changed the bottom axis in the photocurrent plot (Figure 4) to electric field. Based on our calculations, the critical field of type-I to type-II transition is 0.16 V/nm via PL in Device 1, 0.15 V/nm via photocurrent in Device 2, 0.15 V/nm via PL in Device 3, and 0.135 V/nm via photocurrent in Device 4. Overall, the PL and photocurrent results give similar critical electric fields.

Comment: The authors presented in the SI some characterizations which were not mentioned in the main text. Please, add a proper indication. Moreover, the authors mentioned details of measurements in the SI which were not referenced in the main text “i.e., V_{sd} was applied to the WS_2 flake and current was measured from MoSe_2 ”. This was not mentioned in the main text. Please move it to the main text since this helps the understanding of the device.

Response: We thank the reviewer for pointing out these shortcomings in our writing. In the revised manuscript, we have added indications to refer to the related sections or figures in the Supplementary Information. We have also added “ V_{sd} is applied to the WS_2 flake and current is measured from MoSe_2 (Fig. 3a)” in Page 4 of the manuscript.

Reviewer #3 (Remarks to the Author):

Comment: The manuscript presents research work that reveals the switch between the type-I and type-II band alignments in MoSe₂/WS₂ heterobilayers. By applying a vertical electric field, luminescence and photocurrent measurements show that the MoSe₂/WS₂ bilayer can be switched from a type-I band alignment to a type-II band alignment. The authors further developed a phenomenological model to show that the highly nonlinear photocurrent, according to the source-drain electric field, is due to the transition of band alignments and substantial changes in exciton lifetime. The work utilizes the unique layer structures of van der Waals (vdW) materials to achieve the electric-field tunable band alignment, which is challenging in non-vdW materials. This result can be useful for new optoelectronic device applications.

Regarding the current version of the manuscript, I have a few questions:

Response: We thank the reviewer for reading our manuscript and the nice summary. Below we will give our point-to-point reply to the comments. We have revised our manuscript and also added new contents and re-organized the Supplementary Information. In the revised manuscript, we have highlighted the relevant changes in red color for the convenience of reviewers.

Comment: I am not sure about the type-I band alignment in intrinsic MoSe₂/WS₂ heterobilayers. I noticed that there were a few calculations (e.g., Nano Lett. 2016, 16, 7, 4087–4093) showing that it is a type-II band alignment. This manuscript confirms that the CBM is from MoSe₂, which is not enough to confirm the overall type-I band alignment. Moreover, the electron-hole binding energies for intralayer and interlayer excitons are different, and it is questionable to extrapolate the band alignment from the exciton energies in Figure 2c. It is better to employ theoretical calculations of the band structure and excitons to support the claims of this work.

Response: We thank the reviewer for raising this concern. Although some theories may predict a type-II band alignment for the MoSe₂/WS₂ heterobilayer, the prediction may not be accurate due to the simplicity of the theory and the existence of charge transfer, interfacial coupling, change of dielectric environment in the actual heterobilayer samples. Therefore, it is more reliable to determine the band alignment from the experimental data. The MoSe₂/WS₂ heterobilayer has a small interlayer band offset in the conduction band minima (CBM) but a large band offset (~300 meV) in the valence band maximum (VBM). The band alignment type is therefore determined by the CBM offset.

Since our PL data (Fig. 2) shows dominant intralayer exciton emission at zero interlayer electric field, it is clear that both electrons and holes reside in the MoSe₂ layer of the heterobilayer. The major obstacle to clarify the band alignment type (Type I or Type II) is the different binding energies of the intralayer and interlayer excitons. These binding energies can be estimated in the following way.

First, we estimate the binding energy ratio between intralayer and interlayer excitons. In our previous work on MoSe₂/WSe₂ heterobilayer [see Supplementary Information of *Nature* 594, 46–50 (2021)], our theorist calculated the binding energies of the interlayer exciton (114 meV) and intralayer WSe₂ exciton (152.6 meV). The ratio between them is $114/152.6 \approx 75\%$. Second, we estimate the reduction of intralayer exciton binding energy in heterobilayer compared to in monolayer. According to *Phys Rev B* 99, 205420 (2019), the intralayer exciton binding energy in

BN-encapsulated monolayer WSe₂ is 172.1 meV. The additional screening in the heterobilayer therefore reduces the binding energy of the intralayer exciton by $152.6/172.1 \approx 90\%$.

Figure R4 | Estimation of band offset in the MoSe₂/WS₂ heterobilayer. **a**, Schematic of the band alignment. A band offset of 42 meV is obtained from the binding energies of intralayer A_{Mo} exciton (191 meV) and interlayer IX exciton (143 meV) and the A_{Mo} - IX separation (90 meV). **b**, The electric-field-dependent PL map from Figure 2c. A 90 meV energy separation is extrapolated between A_{Mo} and IX at zero field.

It is reasonable to assume similar binding energy ratios in the case of MoSe₂/WSe₂ heterobilayer. We previously determined the intralayer exciton binding energy to be 212.5 meV in BN-encapsulated monolayer MoSe₂ [see Supplementary Information of *Nat Commun* 12, 6131 (2021)]. By using the 90% ratio, the intralayer MoSe₂ exciton binding energy is $212.5 \times 90\% \approx 191$ meV in a MoSe₂/WS₂ heterobilayer due to the additional screening. By using the 75% ratio, the interlayer exciton binding energy is $191 \times 75\% \approx 143$ meV (see Fig. R4a). The binding energy difference between intralayer and interlayer excitons is therefore $191 - 143 = 48$ meV.

By using linear extrapolation in our PL map of MoSe₂/WSe₂ heterobilayer (Fig. R4b or Fig.2c), we find a 90-meV separation between the intralayer and interlayer excitons at zero field. After considering the difference (48 meV) of the intralayer and interlayer exciton binding energies, the conduction band offset is $90 - 48 = 42$ meV (Fig. R4a) That is, the WS₂ CBM is 42 meV higher than the MoSe₂ CBM in our heterobilayer sample. This confirms the type-I band alignment of our WS₂/MoSe₂ heterobilayer.

In the revised manuscript Page 4, we have added, “Considering the different exciton binding energies between A_{Mo} and IX , we further estimate that the WS₂ CBM resides ~ 40 meV above the MoSe₂ CBM ...”. We also added Section 3 in the Supplementary Information to explain our estimation. We thank the referee for the good questions that help us improve our manuscript.

Comment: The field dependence of the lifetime of excitons shown in Figure 4d is interesting. Can the authors convert the V_{sd} to the value of an electric field? I hope to know if it agrees with the crucial field (0.16 V/nm) observed in luminescence measurements.

Response: We thank the reviewer for this important question. We have calculated the electric fields in both PL and photocurrent experiments (see Section 2 in the revised Supplementary Information), and changed the bottom axis in the photocurrent plot (Figure 4) to electric field. Based on our calculations, the critical field of type-I to type-II transition is 0.16 V/nm via PL in Device 1, 0.15 V/nm via photocurrent in Device 2, 0.15 V/nm via PL in Device 3, and 0.135 V/nm via photocurrent in Device 4. Overall, the PL and photocurrent results give similar critical electric fields.

Comment: The explanation of the laser power dependence is not clear to me. It seems that higher power introduces more significant nonlinear curves in Figure 4a. Does it affect the lifetime of intralayer or interlayer excitons?

Response: We thank the referee for pointing out the lack of explanation in our manuscript. It is true that higher laser power introduces more significant nonlinear curves in Fig. 4a. It also shortens the exciton lifetime according to our rate equation:

$$dN/dt = -(\alpha + \beta)N - \gamma N^2$$

Here N is the exciton density; α is the carrier extraction rate; β is the first-order electron-hole recombination rate; γ is the decay rate due to exciton-exciton annihilation or Auger processes. When the exciton density N increases at increasing laser power, the nonlinear biexcitonic decay term ($-\gamma N^2$) becomes more important. This leads to faster exciton decay and photocurrent suppression.

In the revised manuscript Page 6, we addressed this issue by writing, “*When exciton density N increases at increasing laser power, the nonlinear decay term ($-\gamma N^2$) becomes more important, leading to faster exciton decay and photocurrent suppression.*”

REVIEWERS' COMMENTS

Reviewer #1 (Remarks to the Author):

The authors amended their work sufficiently to address remaining points raised by the reviewers. The manuscript can be considered for publication at the current stage.

Reviewer #2 (Remarks to the Author):

I think the authors have adequately responded to the reviewer's concerns. The manuscript is much improved and I recommend publication in its current form.

Reviewer #3 (Remarks to the Author):

The reply and revised manuscript well answered and addressed my questions. Therefore, I recommend it to be published in Nature Comm.

RESPONSE TO REVIEWERS' COMMENTS

Reviewer #1 (Remarks to the Author):

The authors amended their work sufficiently to address remaining points raised by the reviewers. The manuscript can be considered for publication at the current stage.

Response: We thank the reviewer for the positive comment. We greatly appreciate the reviewer's time in helping us improve our manuscript.

Reviewer #2 (Remarks to the Author):

I think the authors have adequately responded to the reviewer's concerns. The manuscript is much improved and I recommend publication in its current form.

Response: We thank the reviewer for the positive comment. We greatly appreciate the reviewer's time in helping us improve our manuscript.

Reviewer #3 (Remarks to the Author):

The reply and revised manuscript well answered and addressed my questions. Therefore, I recommend it to be published in Nature Comm.

Response: We thank the reviewer for the positive comment. We greatly appreciate the reviewer's time in helping us improve our manuscript.